# Promising New Material for Food Packaging: An Active and Intelligent Carrageenan Film with Natural Jaboticaba Additive

**DOI:** 10.3390/foods11060792

**Published:** 2022-03-09

**Authors:** Luisa Bataglin Avila, Elis Regina Correa Barreto, Caroline Costa Moraes, Marcilio Machado Morais, Gabriela Silveira da Rosa

**Affiliations:** 1Engineering Graduate Program, Federal University of Pampa, 1650, Maria Anunciação Gomes de Godoy Avenue, Bagé 96413-172, Brazil; luisabataglinavila@gmail.com; 2Chemical Engineering, Federal University of Pampa, 1650, Maria Anunciação Gomes Godoy Avenue, Bagé 96413-172, Brazil; elisrbarretoeq@gmail.com (E.R.C.B.); marciliomorais@unipampa.edu.br (M.M.M.); 3Graduate Program in Science and Engineering of Materials, Federal University of Pampa, 1650, Maria Anunciação Gomes de Godoy Avenue, Bagé 96413-172, Brazil; caroline.moraes@unipampa.edu.br

**Keywords:** biopolymer, pH indicator, antimicrobial, antioxidant, phenolic compounds, anthocyanins

## Abstract

This research focused on the development of active and intelligent films based on a carrageenan biopolymer incorporated with jaboticaba peels extract (JPE). The bioactive extract was obtained by maceration extraction and showed high concentrations of total phenolic content (TP), total anthocyanin (TA), cyanidin-3-glucoside (Cn-3-Glu), antioxidant activity (AA), and microbial inhibition (MI) against *E. coli*, being promising for use as a natural additive in food packaging. The carrageenan films were produced using the casting technique, incorporating different concentrations of JPE, and characterized. The results of the thickness and Young’s modulus of the film increased in the films supplemented with JPE and the addition of the extract showed a decrease in elongation capacity and tensile strength, in water vapor permeability, and a lower rate of swelling in the water. In addition, the incorporation of JPE into the polymeric matrix promotes a change in the color of the films when compared to the control film and improves the opacity property. This is a positive effect as the material has a UV-vis light barrier which is interesting for food packaging. The increase in the active potential of the films was directly proportional to the concentration of JPE. The films results showed visible changes from purple to brown when in contact with different pH, which means that films have an intelligent potential. Accordingly, this novel carrageenan based-film incorporated with JPE could be a great strategy to add natural additives into packaging material to obtain an active potential and also an indicator for monitoring food in intelligent packaging.

## 1. Introduction

In recent years, environmental concerns have increased globally, which has promoted research for the development of food packaging using biopolymers with the main advantage of the short biodegradation time compared to petrochemical raw materials [1]. On the other hand, another problem discussed is the waste of food, which, according to Poyatos-Racionero et al. [2], goes beyond ethical and economic issues as it can cause the depletion of natural resources such as water and energy. In this scenario, innovations in food packaging appear as a way to minimize these problems and, for this, new concepts are included to obtain packaging that can extend the shelf life of products and ensure food safety for consumers [3,4]. Thus, to develop so-called “active and intelligent packaging”, it is necessary to use additives that can be synthetic or natural.

In this context, jaboticaba is a Brazilian berry belonging to the *Myrtaceae* family that includes genera such as *Plinia* and *Myrciaria*. This fruit is notoriously known for being a rich source of phenolic compounds, especially anthocyanins [5,6]. The species of this fruit are cultivated throughout the country because it adapts in several Brazilian biomes such as Cerrado, Caatinga, Atlantic Forest, Amazon Forest, and Pampa. Moreover, the industrial process of this fruit generates by-products, such as peels and seeds, which have a higher phenolic content than the pulp and are normally discarded in the environment [5]. Although the peels are considered a residue, their extracts highlight their antioxidant, anti-inflammatory, and antimicrobial properties [7]. This suggests that the peel presents itself as a potential and sustainable alternative to the use of synthetic additives.

Considering the challenges in the packaging industry regarding environmental concerns, efforts are made to seek new polymeric sources to minimize the environmental impact. In this sense, there is a wide variety of biopolymer resources, such as polysaccharides, proteins, lipid materials, and mixtures which have been used in the development of renewable and biodegradable films [8]. In this scenario, κ-carrageenan, a polysaccharide obtained by extracting red algae, stands out in the production of polymeric films or hydrogels for food packaging due to being a renewable material, biodegradable, biocompatible, and with good film formation properties [8,9]. The use of this biopolymer is part of the circular bioeconomy concept, as it can be obtained through the use of marine biomass from the seafood processing industry, for example [10].

However, poor barrier and mechanical properties of carrageenan films are known, especially due to the high hydrophilicity. For this reason, some authors have studied the effect caused by the addition of substances such as plant extracts on film properties Liu et al. [11]. Farhan and Hani [12], in their study, developed a carrageenan active edible film for chicken breast packaging. On the other hand, Chi et al. [13] produced an intelligent film using carrageenan as a polymer and grape skin as an additive. Liu et al. [11] focused their efforts to make a carrageenan packaging film incorporated with mulberry polyphenolic extract with pH-sensitive and antioxidant properties and Martiny et al. [14] developed a biodegradable film based on κ-carrageenan activated with olive leaves extract for antimicrobial food packaging.

In another work, the authors of the present study published an article along the same lines, however evaluating another extraction technique. Furthermore, the mentioned article only considers the active potential of the material [15]. In view of the above, this manuscript aims to explore a new possible application of jaboticaba peel extract. This feat is accomplished through the study and development of a film based on carrageenan biopolymer for food packaging that, in addition to its active potential, can also act as intelligent packaging.

## 2. Materials and Methods

### 2.1. Reagents

Jaboticaba fruits, fully mature, were collected from a farm located at Santa Maria, Rio Grande do Sul, Brazil in November 2019 to be processed in the following year. Hydrochloric acid, carrageenan, glycerol, 2,2-diphenyl-1-picrylhydrazyl (DPPH), Folin Ciocalteu reagent, anhydrous sodium carbonate, and gallic acid were of analytical grade. Water, acetonitrile, formic acid, and cyanidin-3-glucoside were of HPLC grade. For the antimicrobial analysis: Nutrient and Müller-Hinton broth were used. All reagents were purchased from Sigma Aldrich (St. Louis, MO USA). The bacteria strain used in the antimicrobial experiment was *Escherichia coli* ATCC 11229 supplied by The Fiocruz–Oswaldo Cruz Foundation, Rio de Janeiro, Brazil.

### 2.2. Extract Preparation

Firstly, the peels were manually separated from the pulp. After that, the peels were sanitized with a commercial solution of 2% sodium hypochlorite and rinsed in sterilized distilled water, then dried in a freeze-dryer (Terroni, LS3000, São Carlos, Brazil) for 48 h. After this process, the freeze-dried peels were reduced to a fine powder by an analytical mill (IKA, A11, Darmstadt, Germany) and sieved (Metallic mesh size 60, Metallurgical Industry Bertel, Caieiras, Brazil) to select particles with less than 0.272 mm.

Then, the jaboticaba peel extract was obtained by maceration according to the optimal condition described by Avila et al. [16] using acidified water (pH 1) as solvent and extraction temperature of 88 °C for 1 h. These parameters were defined according to an experimental design that evaluated the parameters of extraction temperature and solvent pH for the recovery of total phenolic compounds and total anthocyanin from freeze-dried jaboticaba peels. The extraction step was performed in a water-bath (SOLAB—SL 157, Piracicaba, Brazil) under constant agitation. The extract was subjected to vacuum filtration, freeze-dried (Terroni, LS3000, São Carlos, Brazil), and stored in vacuum-sealed bags.

### 2.3. Extract Characterization

Total phenolic content (TP) was quantified by the spectrophotometric method adapted from proposed by Singleton and Rossi [17], was used Folin-Ciocalteu reagent and standard curve with different concentrations (72–1800 mg L^−1^) of gallic acid. The method is based on a colorimetric oxidation/reduction reaction which has its intensity read in a spectrophotometer and related to the concentration of phenolic compounds. All measurements were made in triplicate and the results were expressed as milligrams of gallic acid equivalent (GAE) per gram of dried matter.

The antioxidant activity (AA) was determined with the method developed by Brand-Williams, Cuvelier & Berset [18]. This method is based on the color change that occurs when the DPPH radical is captured by the antioxidant present in the sample. The color change caused by the oxidation/reduction reaction is then read in a spectrophotometer. DPPH (6 × 10^−5^ M) was prepared with the DPPH standard and methanol. The absorbance of blank, as well as solutions with extracts, was measured using a spectrophotometer (UV 755B, EQUILAM, Diadema, Brazil) at 517 nm. The analysis was conducted in triplicate and the results were expressed in percentage of free DPPH radical scavenging.

Total anthocyanins (TA) were estimated through the spectrophotometer method with some modifications [19,20,21]. The absorbance of the extract solution was measured at a wavelength of 520 nm, using a spectrophotometer. The standard curve with concentrations ranging from 5 to 100 mg L^−1^ was used to measure the concentration of anthocyanin in the jaboticaba peels, and the results were expressed as mg of cyanidin-3-glucoside (Cn-3-Glu) equivalent per 100 g of dried matter.

The quantitative analyses of Cn-3-Glu in JPE were performed by high-performance liquid chromatography (HPLC), using an Agilent 1260 Infinity Series (Santa Clara, CA, USA), equipped with a variable wavelength detector (VWD). The isocratic mobile phase consisted of water/acetonitrile/formic acid solvents (80/10/10 *v*/*v*/*v*). The separation was conducted at 30 °C using a reversed-phase Discovery column (Supelco, Bellefonte, PA, USA) RP C18 (5 μm, 25 cm × 4.6 mm). The injection volume was set at 20 μL, and the flow rate of the mobile phase was 1 mL·min^−1^. The extract obtained was filtered through a 0.45 mm syringe filter and injected directly into the HPLC. The detector was set at 530 nm, and each sample run was conducted for 10 min. The concentrations of Cn-3-Glu in the extracts were quantified using a standard curve, with concentrations ranging from 500 to 4000 mg L^−1^ with characteristic peak presenting retention time between 3 and 4.5 min. The extraction yields of Cn-3-Glu were expressed in mg 100 g^−1^ of dried matter.

Microbial inhibition (MI) against *E. coli* was obtained following the micro-dilution method adapted to that described in the standard M07—A10 of Clinical and Laboratory Standards Institute (CLSI) [22]. The sample was tested at the concentration of 50 mg·mL^−1^. For the analysis, 135 μL of extract, 145 μL of sterile Muller-Hinton broth, and 20 μL of the *E. coli* culture were added in a 96-well microliter plate. A control (inoculum without extract) was included on a microplate. The microplate was incubated at 35 °C for 16 h. The absorbance was read before the incubation period (0 h) and after the incubation period (16 h) at 630 nm wavelength (OD630) using a microplate reader (Celer-Polaris, Belo Horizonte, Brazil). The contents of the wells were mixed before reading the absorbances and they are expressed as a percentage of growth inhibition. Each experiment was repeated three times.

### 2.4. Preparation of Carrageenan Based-Film

Carrageenan based-films were developed by a casting method according to the methodology proposed by Avila et al. [15] which consists of dissolved 0.5 g of carrageenan in 30 mL of distilled water being heated at 70 °C at 15 min under constant agitation (110 rpm) using a hot plate and magnetic stirrer (QUIMIS—Q261M23, Diadema, Brazil). The glycerol was used as a plasticizer and added into a carrageenan solution, at a concentration of 60% *w*/*w* based on the weight of the polymer, during agitation. Thereafter, the freeze-dried jaboticaba peel extract (JPE) was added to the already cooled solution at concentrations of 50 and 100% *w*/*w* based on the weight of the polymer. After that, the polymeric solutions were molded in the polystyrene Petri dishes (90 mm diameter). Casting and drying were carried out at 40 °C using a convective dryer. After drying, the films were then manually peeled off from plates and conditioned in a relative humidity of 50% and room temperature for 48 h for further analysis. A biodegradable film of carrageenan without the addition of jaboticaba peels extract was also produced at the same conditions and called CAR-control. The biodegradable film of carrageenan with 50 and 100% of JPE is called CAR-50% JPE and CAR-100% JPE, respectively.

### 2.5. Carrageenan Film Characterization

#### 2.5.1. Thickness

The thickness (*e*) was measured at ten random locations on the film by using a digital micrometer (Insize-IP65, São Paulo, Brazil). The accuracy of the micrometer was 0.001 mm.

#### 2.5.2. Color

The color of the films was evaluated using a spectrophotometer (Konica Minolta, CM-2600D, Tokyo, Japan). The color values of the film, including *L** (lightness/brightness), *a** (redness/greenness), and *b** (yellowness/blueness), were measured and the total color difference (Δ*E**) was calculated using Equation (1):Δ*E** = [(*L** − *L_s_**)^2^ + (*a** − *a_s_**)^2^ + (*b** − *b_s_**)^2^]^(1/2)^(1)
where, the sub-index *s* represents the parameters of the film control, without JPE. The color difference was expressed in %. The film opacity was determined by the methodology used by [23]. Absorbance was measured at 600 nm using a spectrophotometer (UV 755B, EQUILAM, Brazil).

#### 2.5.3. Mechanical Properties

The mechanical properties of the films were analyzed by measuring the tensile strength (TS), elongation percentage at break (E) and Young’s modulus using a texture analyzer (STABLE MICRO SYSTEM—TA.XT.plus, Surrey, UK) according to the ASTM Standard D882-18 [24]. Samples were clamped and deformed under tensile loading using a 50 N load cell with an initial grip separation of 25 mm and cross-head speed of 50 mm min^−1^.

#### 2.5.4. Water Vapor Permeability (WVP)

The water vapor permeability of films was determined gravimetrically using the ASTM E 96/E 96M-16 [25] method. The film samples were sealed in permeation cells containing anhydrous calcium chloride and stored in desiccators with a relative humidity of 50%. The permeation cells were weighed on the first day and after seven days.

#### 2.5.5. Swelling Index (SI)

The SI was analyzed using the methodology adapted from Bunhak et al. [26]. The films were cut into samples of 0.001 m^2^ and inserted in a drying oven at a temperature of 105 °C, where they remained for a period of approximately 24 h. The films were weighed on an analytical scale to determine the dried weight and immediately immersed in a medium containing distilled water for 10 min. After the time of immersion, it was removed from the environment where they were immersed and reweighed. The kinetic characteristics of the swelling of the films were quantified and expressed as a percentage.

#### 2.5.6. Bioactive Properties

To determine the TP and AA in the films, the same methodology was used, which was applied for JPE (Section 2.3) and according to methodology proposed by Perazzo et al. [27] (with some modifications), the use of 0.1 g of the film was standardized.

#### 2.5.7. FTIR-ATR Analysis of the Films

Infrared spectroscopy with attenuated total reflectance Fourier transform (FTIR-ATR) was used to identify the functional groups present in the films, as well as the interaction between the extract and the polymer. The attenuated total reflectance Fourier transform infrared spectroscopy (FTIR–ATR) was used to investigate the chemical composition of the films, as well as the interaction between the extract and the polymer. A Perkin-Elmer spectrometer (UATR Two), in the range of 400 cm^−1^ to 4000 cm^−1^, was used with 32 scans per spectrum and with a resolution of 4 cm^−1^. For the analysis of the films, samples were cut into small squares and then inserted into the sample portal of the FTIR-ATR device to obtain the spectra to be analyzed.

#### 2.5.8. Determination of the pH–Sensitive Property

The sensitivity of films to pH changes was determined according to Luchese et al. [28] with some modifications. Samples of the film (2 × 2 cm^2^) were immersed in different buffer solutions ranging from pH 1 to 12 for 20 min. Then, the samples were dried in an oven at 40 °C for approximately 30 min and the color changes were determined by a digital colorimeter as described in Section 2.5.2 but in relation to the original films (CAR-50% JPE and CAR-100% JPE that were not immersed into a buffer solution).

#### 2.5.9. Statistical Analysis

Experimental data were analyzed by Statistica® software, version 10.0 (SAS Institute, Cary, NC, USA). The mean comparisons were carried out by Tukey test and *t*-test were applied for determining significant differences at 95% significance level.

## 3. Results and Discussion

### 3.1. Bioactive Properties of Extract

According to the extract characterization the values obtained for JPE, at optimal condition (88 °C and solvent at pH 1) were 199.34 ± 2.13 mg_GAE_.g^−1^ (d.b.), 81.00 ± 0.72%, 1458.11 ± 0.01 mg 100 g^−1^ (d.b.), 718.12 ± 25, 86 mg 100 g^−1^ (d.b.) phenolic (TP), antioxidant activity (AA), total anthocyanin (TA), and cyanidin-3-glucoside (Cn-3-Glu) respectively.

Thus it is possible to observe the presence of high antioxidant activity due to the high phenolic contents especially for anthocyanins (mainly Cn-3-Glu). This fact is explained by Schreiber et al. [29] and reinforced by Barros et al. [30], who relate the antioxidant activity with phenolic compounds capacity to inhibit oxidative reactions. Palozi et al. [7] found a concentration of 181.42 ± 3.67 mg_GAE_ g^−1^ for jaboticaba peel extract using water as a solvent and accelerated solvent extraction methodology. Alara et al. [31] studied the effect of the parameters involved in microwave-assisted extraction on the recovery of these compounds and reported values of TP and AA of 102.6 ± 1.2 mg_GAE_ g^−1^ and 61.15 ± 0.93%, respectively. Other vegetable wastes are reported in the literature as a source of antioxidant compounds. Rosa et al. [32] evaluated the effect of different extraction methods to recovery bioactive compounds of olive leaves and found, for aqueous extract obtained by maceration, a value of 67.25 ± 0.03% for antioxidant activity. Hacke et al. [33] studied the effect of different solvents and different extraction conditions (temperature and time) in the antioxidant compounds from jaboticaba seeds and reported values between 11.02 ± 0.30% and 82.79 ± 0.50%. These results emphasize the importance of the high values obtained in this study. The total anthocyanin in jaboticaba peel is also reported in the literature, as Lenquiste et al. [34] found a value of 404.56 ± 35.85 mg 100 g^−1^ JPE using hydroethanolic solution as a solvent and Quatrin et al. [35] reported 1153 mg·100 g^−1^ for JPE obtained using methanol/water/formic as solvent. According to Qin et al. [36], phenolic compounds are promising to use as active food packaging and since the process of food deterioration is usually accompanied by changes in pH, the incorporation of anthocyanin (a type of phenolic compounds) in films also shows promise for use as intelligent pH indicators to monitor the freshness of food.

In comparison with the results described in this study, some authors have reported the presence of cyanidin-3-glucoside as a majority compound in the jaboticaba extract. Wu et al. [37] reported a concentration of 298 ± 1.73 mg 100 g^−1^ for jaboticaba fruit extract, on the other hand, while Inada et al. [38] showed higher results, with 1261 mg 100 g^−1^ using methanol as solvent and exhaustive extraction. The high value described by the last authors may be related to the extraction method and the type of solvent used. According to the literature, methanol promotes better results in the recovery of bioactive compounds [39]. However, for future applications in food products, the use of water as a solvent has the advantage of being safer and more environmentally friendly when compared to organic solvents.

The microbial inhibition of JPE, at different concentrations, against *E. coli* was evaluated, as shown in Table 1.

From the results presented in Table 1, it was possible to observe the microbial inhibition of JPE at all concentrations analyzed against *E. coli*. Furthermore, no significant difference was observed in the inhibition values when the extract concentration was reduced. This fact indicates that even low concentrations of JPE are able to inhibit the growth of the microorganism *E. coli*. The microbial inhibition of the extract is related to its composition, which is rich in phenolic compounds. According to Mandal et al. [40], these compounds exhibit several important properties and among them is antimicrobial activity. Many authors have reported in their studies this property of jaboticaba fruit and leaves [41,42]. Girennavar et al. [43] evaluated the microbial inhibition of Marsh white grapefruit juice and reported a value of 10.6% against *E. coli*. Thus, the characteristics of the extract presented above are promising for the application as a natural additive in active and intelligent food packaging.

### 3.2. Carrageenan Film Characterization

The novel carrageenan based-film incorporated with JPE was developed and characterized and shown to be homogeneous and uniform, as can be seen in Figure 1.

Table 2 shows the results of biodegradable film characterization.

From the data presented in Table 2, the film thicknesses were in the range of 0.039 to 0.055 mm and only when applied the highest JPE concentration there was a significant difference between the samples. This fact can be attributed to the high concentration of anthocyanins in the extract. Sun et al. [44] and Wang et al. [45] explain that large amounts of hydroxyl groups present in the anthocyanins compounds can act as bridges and bond strongly to the polymer, forming a network structure through intermolecular interactions, thus increasing the thickness of the film. In order to explain the increase in film thickness, Rasid et al. [46] and Wu et al. [47] explain that polyphenolic compounds could fit into carrageenan matrix and established cross-links through hydrogen bond or hydrophobic interaction with reactive groups of carrageenan and promotes a film network with a decreased free volume of the polymeric matrix.

The elongation at break was significantly affected by the addition of JPE in the polymeric matrix and this effect promoted a reduction in this parameter which was inversely proportional to the increase in the concentration of extract in the films. The same was observed in the tensile strength. Chi et al. [13] observed the same behavior in carrageenan films incorporated with grape skin powder and attributed this to the presence of the molecules which interact and form defect points and points of concentration of stress in the films. Avila et al. [15] reported similar results and the same trend when JPE obtained by microwave-assisted extraction was added into the carrageenan matrix, with values of tensile strength of 10.69 ± 1.61 MPa for carrageenan film control and 6.08 ± 0.33 MPa for carrageenan film with JPE. Sun et al. [44] reported a reduction in tensile strength of carrageenan films when *Prunus maackii* extract at the concentration of 8% (*w*/*w*) was added, from 15.35 ± 0.40 to 10.78 ± 0.17 MPa. Roy and Rhim [8] reported similar values to elongation break, 4.4 ± 0.3% for carrageenan film, and Liu et al. [11] reported values from 14.49 ± 1.78% for carrageenan film control to 8.59 ± 2.19% for carrageenan film with 4% of the mulberry polyphenolic extract.

The Young’s modulus, or elastic modulus, of the films, had no significant effect by adding JPE. However, although not significant, there was a reduction in the values obtained for this parameter with the addition of the JPE. Carissimi et al. [48] observed the same behavior when adding microalgae to the starch film matrix and attributed this effect to hydrogen-bridge interactions between the polymer and extract that cause a decrease in molecular motility. In this case, the extract acts as an anti-plasticizer, reducing the flexibility of the film. The results shown in Table 2 are in agreement with those reported by Dyshlyuk et al. [49], who found values of 80.1 ± 8.0 and 130.8 ± 13.1 MPa for carrageenan films with different compositions.

The WVP did not show a significant difference by the smaller addition of JPE nevertheless, the increase of JPE concentration into polymeric matrix caused a significant reduction in this parameter. Liu et al. [11] reported the same trend when mulberry polyphenolic extract was added into the carrageenan matrix at different concentrations. The same author reported values in a range of 7.83 to 3.86 × 10^−11^ g·m^−1^·Pa^−1^·s^−1^ of carrageenan control and carrageenan with different concentrations of extract (1, 2 and 4%). Martiny et al. [14] also reported the same trend when olive leaves extract was added into the carrageenan matrix and attributed this fact to the reduction in mass diffusion. In general, the difference in WVP is related to material composition. In this case, the addition of the extract may have promoted an interaction between phenolic compounds and biopolymer that fills the carrageenan matrix. This fact creates a difficulty in mass diffusion and consequently a reduction in water vapor permeability. Siripatrawan and Harte [50] studied the incorporation of green tea extract into chitosan matrix for active film and observed the same behavior.

When comparing the values of carrageenan-based film with values of chitosan (a water-insoluble biopolymer), carrageenan films have greater water vapor permeability. Kalaycıoglu et al. [51] reported 1.53 to 1.12 × 10^−10^ g·m^−1^·Pa^−1^·s^−1^ for chitosan based-film with and without turmeric extract.

Therefore, the results obtained in this study are positive and promising for applications in food packaging, as they imply better conditions for food preservation.

The swelling index for all films was high due to the hydrophilicity of κ-carrageenan, which is very hydrophilic, absorbing water and swelling rapidly [52]. The addition of JPE at a concentration of 100% induced a significant reduction in its swelling index. This is possibly related to the greater thickness of films containing JPE, which inhibited the water molecules from moving in and out of the films [53]. In accordance with what was previously described for WVP, the swelling index showed a significant difference between the control film and the films with different concentrations of JPE caused by interaction between extract and the polymeric matrix. Park et al. [54] explain that the filling effect caused by the addition of extract can promote a reduction in the hydrophilicity of the material and a consequent reduction in swelling index.

Regarding the color of the films, all parameters showed significant differences. In this sense, the addition of the extract promoted a clear increase in reddish and yellowish tones (increase in *a** and *b**). These differences between parameters *L**, *a**, and *b**, of the films with and without JPE, resulted in a difference of 79.99% and 96.94% for the films with 50 and 100% of JPE, respectively, in relation to the control film. A similar result was obtained by Avila et al. [15], who reported a difference in film color of 96.50% for a film containing JPE in comparison with carrageenan film without JPE. This result may be due to the composition of the extract, since the phenolic compounds and anthocyanins present in the jaboticaba peels are responsible for their color, and may vary from purple, red, or violet [55]. Besides that, according to Riaz et al. [23], the changes in optical properties of films with the addition of natural extracts, such as decreasing the lightness and film transparency values and increasing the reddish and yellowness values, can be attributed to the addition of antioxidant and antimicrobial compounds.

In line with this, the opacity increased with the increase of the JPE into the films and there was no significant difference between the different concentrations of extract used. This result suggests that films with extract showed higher UV–vis light barrier property as described by Qin et al. [36], who evaluated the effect of the addition of *L*. *ruthenicum* anthocyanins into cassava starch films and observed the same trend. Wu et al. [56] also evaluated the opacity of the films when adding natural extract and reported the behavior described above. The increase of black rice bran anthocyanin increased the opacity of the films.

The films were evaluated for their bioactive properties, as phenolic compounds and antioxidant activity, and the results are shown in Table 3. 

From the results shown in Table 3, it is possible to observe that the addition of JPE into biodegradable film resulted in films with good antioxidant properties, as expected. The inclusion of JPE induced a significant increase in the bioactive compounds and this fact is related to the concentration of phenolic content into the films which followed the same trend. Shojaee-Aliabadi et al. [57] reported the antioxidant activity of carrageenan film with 2% of *Zattaria multiflora* Boiss to be around 60%, a value very close to that found in the present study. In comparison with the bioactive properties of the extract, the films showed a reduction in total phenolic content of up to 39%. In line with this, the antioxidant activity of the films presented a decrease when compared with the antioxidant activity of the extracts. This reduction was up to 48. These results are related to the stability of polyphenols, which can be easily degraded. Paini et al. [58] mention some factors that can cause the degradation of phenolic compounds, such as exposure to light, oxygen, temperature, and enzymatic activities.

The pH-sensitive property of the films was evaluated by the color change of the films when submitted to different pH buffer solutions pH changes. The results of color changes of films are presented in Figure 2 and Figure 3, showing the appearance for visual examination of the films incorporated with JPE.

The results presented in Figure 2 and Figure 3 show that films had pH-sensitive properties. The color change of the films, when subjected to solutions with different pH, is related to the presence of the anthocyanins in the polymeric matrix. At acidic pH (1 and 2) the films tend to maintain their original color which can be explained by the stability of anthocyanins in acidic media [59]. As the pH approaches neutrality an alkaline, it was possible to observe that the films obtain a brownish color. The same was observed by Jayakumar et al. [3] who developed Starch-PVA intelligent films with phytochemicals.

The different colors in the films are attributed to the presence of different molecular species of anthocyanins since at pH 1–3 there is a kind of flavilium cation that is light red, at pH 4–6 the species is the pseudo-base of carbinol that is purple and at pH 7–8 the formation of the quinoid-based species occurs, which is bluish purple [60]. Our results suggested carrageenan films with JPE could be used as pH sensors in the food packaging industry. In this sense, fish is an interesting food option to be packaged in this type of packaging, since fish consumption worldwide is increasing. According to the Food and Agriculture Organization of the United Nations (FAO) [61], in the period 1961–2017, the total consumption rate of fish food was higher than that of all other animal proteins, showing an average growth of 3.1%, while animal proteins showed an average growth of 2.1%. Besides that, a food package that monitors the freshness of this type of food based on pH variation is very interesting, as fish pH is an important health parameter. According to Brazilian legislation [62], the pH of fresh fish should be less than 7.0.

The functional groups present in the freeze-dried extract and in the carrageenan powder, as well as the chemical interaction between them in the biodegradable films, was investigated by FTIR-ATR analysis and are presented in Figure 4.

The broad band ranging from 3260 to 3300 cm^−1^ refers to the stretch of bonded hydroxyl groups (O-H) and the peak at 2930 corresponds to stretching of C-H bonds. Another peak around 1722 cm^−1^, polymer and films spectra, may be related to the stretch of carbonyl groups (C=O) contained in D-galactose, which is a monomer of carrageenan and CC vibration as reported by [53,63]. According to the same author, the region between 1500 and 800 cm^−1^ corresponds to the region of the fingerprint of carrageenan because it contains certain peaks that characterize this polymer. Based on that, it is possible to observe a band at 1202–1220 cm^−1^ of the ester sulfate groups, a sharp peak at 1015–1036 cm^−1^ which corresponds to glycosidic linkages, a band around 930 cm^−1^ related to 3,6-anhydrogalactose ring, and a band at 778–850 cm^−1^ due to the galactose-4-sulfate. Similar results were reported by Farhan and Hani [64] who evaluated the FTIR spectra of carrageenan and interactions between carrageenan and glycerol. In general, polysaccharides represent structural similarities but may show some differentiation from spectra in the region comprising the C-C and C = O vibrations [59].

In relation to the extract, it is possible to observe a peak at 1630 that can be related to the aromatic ring, which is confirmed by literature [65]. According to what was expected, this signal was less intense in the films containing JPE than the freeze-dried extract. Similar results were obtained by Avila et al. [16], who reported the same behavior in ultrafine fibers produced with different concentrations of freeze-dried jaboticaba peel extract. Comparing the JPE containing films with the control film, it was possible to observe a band shift of 3300 for higher wavenumbers as JPE is added. According to Zhou et al. [66], this fact is related to the interaction between anthocyanins, present in the extract, and carrageenan matrix. Another change between the control film and extract embedded films is observed in the range around 1262 cm^−1^. This band showed a reduction in wavelength proportional to the concentration of extract in the film. Zepon et al. [67] attributed this effect to the cationic origin of anthocyanins that in pH values promote an electrostatic bond between oxygen (with a positive charge) of anthocyanins and sulfate ester (with a negative charge) of carrageenan. This statement is in agreement with what was observed in the present study since the extract was obtained using an acidified solvent (pH 1).

## 4. Conclusions

The JPE obtained by maceration showed high concentrations of phenolic compounds, anthocyanins, especially cyanidin-3-glucoside (Cn-3-Glu). Such compounds have high antioxidant and antimicrobial activity that demonstrate the high potential of JPE for use as a natural additive. These properties could be transferred to the carrageenan films by adding the extract to the polymeric matrix which proves the possibility of adding value to a residue such as jaboticaba peels. The process to obtain the active films did not provoke a degradation of the bioactive compounds, which were confirmed by FTIR analysis, and bioactive compounds that demonstrated the presence of JPE in the films.

The addition of JPE to the films caused a reduction in the mechanical properties. However, it promoted positive changes, such as a reduction in water vapor permeability (WVP) and swelling index (SI) and increase in opacity. Furthermore, the color of films containing JPE changed when compared to CAR-control. Furthermore, the presence of anthocyanins in the JPE allowed the development of colorimetric indicator films, as they presented color changes when exposed to different pH buffer solutions. The application of this film as food packaging is still a challenge. For this reason, it is intended in future studies to evaluate its application in fish, since the consumption of this type of food is constantly growing. In addition, pH is an important sanitary parameter, and an pH above 7 is considered unfit for consumption.

Based on this, the biodegradable film of carrageenan with the addition of JPE can be considered promising for applications in food packaging with active properties, capable of improving food safety and increasing the shelf life of the product. Therefore, this novel carrageenan based-film incorporated with JPE is promising for use as an indicator in intelligent packaging for monitoring the freshness of packaged foods.

## 5. Patents

Rosa: G.S.; Morais, M.M.; Moraes, C.C.; Avila, L.B.; Barreto, E.R.C.; Martiny, T.R. Filme Inteligente e Ativo à Base de Biopolímero e de Aditivo Natural Obtido do Processamento da Jabuticaba. BR102020026052. Dez., 12, 2020.

## Figures and Tables

**Figure 1 foods-11-00792-f001:**
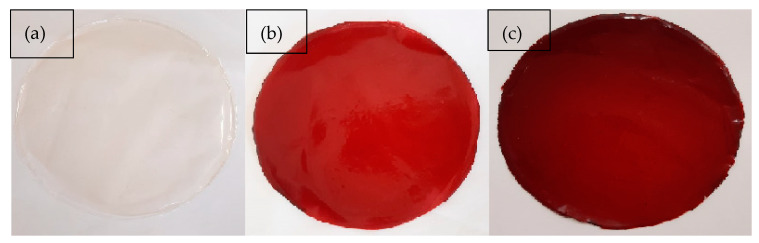
The visual appearance of carrageenan biodegradable films: (**a**) CAR-control, (**b**) CAR-50% JPE and (**c**) CAR-100% JPE.

**Figure 2 foods-11-00792-f002:**
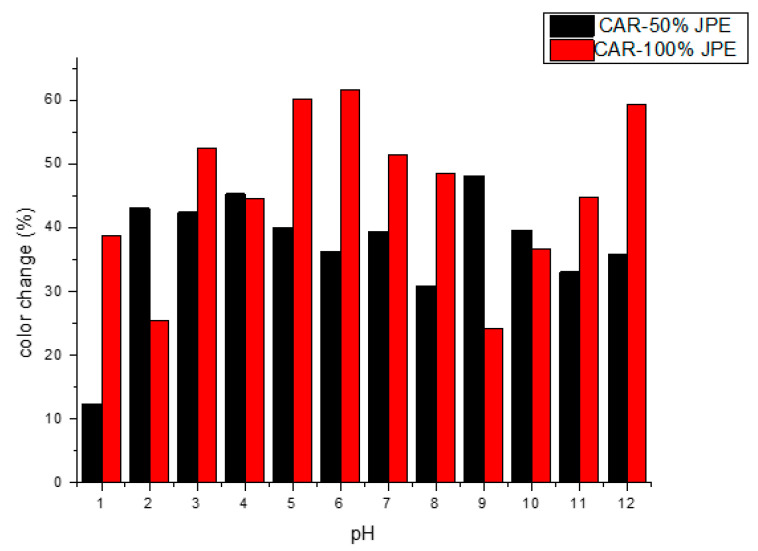
Colorimetric results measured after the sample contact with different buffer solutions and their respective pH value.

**Figure 3 foods-11-00792-f003:**
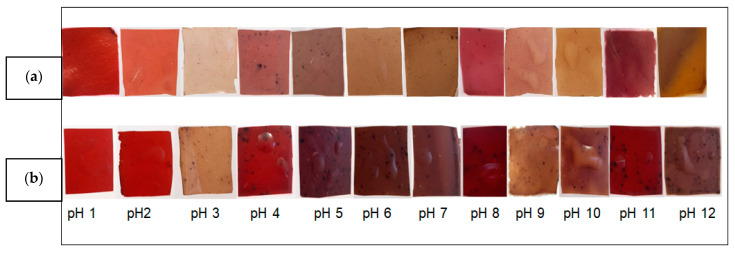
Visual changes in (**a**) CAR–50% JPE and (**b**) CAR–100% JPE films at different pH buffer solutions.

**Figure 4 foods-11-00792-f004:**
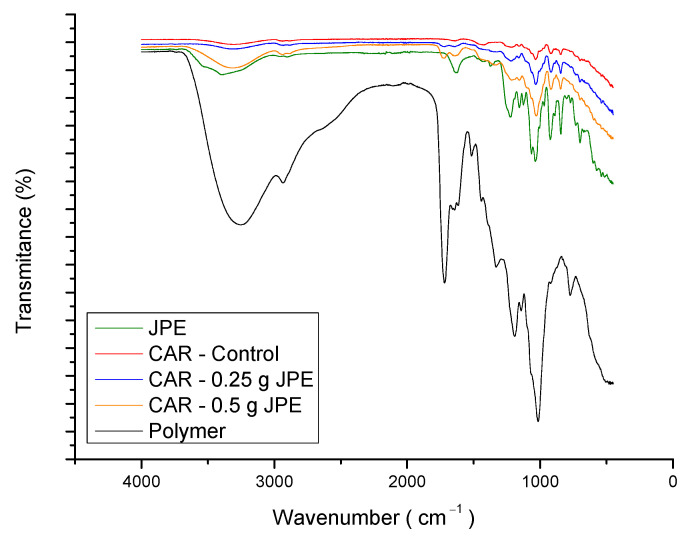
FTIR spectra of JPE, carrageenan powder, and biodegradable films, with and without JPE at different concentrations.

**Table 1 foods-11-00792-t001:** Microbial inhibition of JPE against *E.coli*.

Concentration of JPE (mg mL^−1^)	Microbial Inhibition (%)
50	22.21 ^a^ ± 5.43
16.7	20.18 ^a^ ± 2.73
9.09	12.51 ^a^ ± 5.59

Same letters in the same column indicate that there are no significant differences between samples in Tukey test (*p* < 0.05).

**Table 2 foods-11-00792-t002:** Biodegradable films characterization.

	CAR-Control	CAR-50% JPE	CAR-100% JPE
Thickness (mm)	0.039 ^a^ ± 0.0024	0.042 ^a^ ± 0.0020	0.055 ^b^ ± 0.0018
Elongation at break (%)	10.75 ^a^ ± 2.01	4.61 ^b^ ± 0.10	3.28 ^b^ ± 0.62
Tensile strength (MPa)	7.72 ^a^ ± 0.49	4.03 ^b^ ± 0.57	3.24 ^b^ ± 0.32
Young modulus (MPa)	74.75 ^a^ ± 14.11	87.18 ^a^ ± 10.74	101.57 ^a^ ± 10.44
Water Vapor Permeability (WVP) (g·m^−1^·Pa^−1^·s^−1^)	1.89 × 10^−11 a^ ± 8.40 × 10^−14^	1.80 × 10^−11 b^ ± 5.99 × 10^−13^	1.34 × 10^−11 b^ ± 1.46 × 10^−12^
Swelling Index (%)	95.08 ^a^ ± 1.12	92.02 ^b^ ± 1.51	92.40 ^b^ ± 1.00
*L**	91.96 ^a^ ± 0.22	30.08 ^b^ ± 3.39	21.11 ^c^ ± 0.79
*a**	2.30 ^a^ ± 0.03	56.80 ^b^ ± 1.41	51.00 ^c^ ± 0.87
*b**	−6.80 ^a^ ± 0.31	41.69 ^b^ ± 5.10	37.49 ^b^ ± 3.55
Δ*E* (%)	-	79.99 ± 5.98	96.94 ± 1.70
Opacity (Abs_600nm_.mm^−1^)	4.40 ^a^ ± 0.54	12.18 ^b^ ± 1.16	14.86 ^b^ ± 2.18

Average ± std. deviation (*n* = 10 for thickness, *n* = 9 for swelling index, *n* = 3 for mechanical properties, *n* = 3 for WVP, *n* = 6 for color and *n* = 3 for opacity). Different letters in the same line indicate significant differences between samples in Tukey test (*p* < 0.05).

**Table 3 foods-11-00792-t003:** Bioactive properties of films.

Film	Total Phenolic (TP) (mg_GAE_ g^−1^) (d.b.)	Antioxidant Activity (AA) (%)
CAR-Control	0	0
CAR-50% JPE	121.16 ^a^ ± 40.86	41.84 ^a^ ± 1.59
CAR-100% JPE	140.09 ^a^ ± 2.08	58.91 ^b^ ± 0.55

Average ± std. deviation (*n* = 3 for TP, *n* = 3 for AA). Different letters in the same line indicate significant differences between samples in *t*-test (*p* < 0.05).

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
