# Peer review of "Promising New Material for Food Packaging: An Active and Intelligent Carrageenan Film with Natural Jaboticaba Additive"

_foods, 2022, doi:10.3390/foods11060792_

Round 1

Reviewer 1 Report

This is an original study with a clear focus on sustainable food packaging applications. However, the structure of the manuscript and the discussion need to be highly improved.

- Introduction should be rewritten. The first three paragraphs can be removed or reduced in a single one. Authors should place, instead, a more detail on the use of carrageenan, indicating the advantages and limitations of this biopolymer for packaging applications. This should be related to the concept of Circular Bioeconomy and the valorization of food waste residues to develop biodegradable packaging materials. Reference 20, which deals with a previous study of the group, should be better explained, describing the findings that serve as the basis to develop this new study. Use a more scientific style for the seentence "The group of authors has also published an article about...".

- Describe the equipment used for the extraction process and the conditions in section 2.2. I also suggest to merge sections 2.3 and 2.5.6. Moreover, the selection of compositions should be justified.

- The results section should be divided into subsections, following a logical order, which can be based on that presented in the Experimental part. Further comments on this section: 1) Compare the antioxidant properties of this extract with others obtained from vegetable and fruit wastes to elucidate the potential of these ones; 2) Include a table/graph for the antimicrobial results and improve the discussion; 3) Include and discuss the values of the Young modulus in the mechanical properties; 4) The WVP is not related to the film thickness but to material composition. Values should be compared with other biopolymer films since this material is expected to show a poor water barrier due to its high hydrophilicity; 5) Explain why the extract reduced swelling but did not provide a significant effect on the water vapor performance; 6) The active properties of the films should be compared with the neat extract to determine the effect of processing and release from the film samples; 7) The discussion of the FTIR measurements should be focused on the interaction of the extract with the polysaccharide. 

- In the conclusion section authors can better explain the design of the intelligent packaging material based on pH changes. In other words, what type of food or good can be packaged so that it can present such a large changes on the pH values and then show color variations.  

Author Response

 Promising new material for food packaging: an active and intelligent carrageenan film with natural jaboticaba additive

Author's Response

ATTENTION:

The Edıtor-In-Chıef Measurement

RESPONSES TO EDITORS’S AND REVIEWER’S COMMENTS ON MANUSCRIPT FOODS- 1415889.

Title: Promising new material for food packaging: an active and intelligent carrageenan film with natural jaboticaba additive

First of all, we would like to thank you for making your time available and kindly evaluating our manuscript. Your suggestions and comments have been valuable, thus helping to improve the quality of our manuscript. After a detailed analysis of the comments and questions, as well as the errors and suggestions contained in its opinion, the manuscript underwent some changes. We respectfully acknowledge and appreciate the invaluable contributions of the reviewer. Corrections made are highlighted in red color font. We describe below the treatment given to each of the amendments proposed by the reviewer for resubmission of the manuscript Foods-1415889.

Reviewer #1

#1 This is an original study with a clear focus on sustainable food packaging applications. However, the structure of the manuscript and the discussion need to be highly improved.

RESPONSE: We thank the reviewer for appreciating our effect. The manuscript changed, as suggested, and the modifications are indicated in the sequence.

#1 Introduction should be rewritten. The first three paragraphs can be removed or reduced in a single one. Authors should place, instead, a more detail on the use of carrageenan, indicating the advantages and limitations of this biopolymer for packaging applications. This should be related to the concept of Circular Bioeconomy and the valorization of food waste residues to develop biodegradable packaging materials. Reference 20, which deals with a previous study of the group, should be better explained, describing the findings that serve as the basis to develop this new study. Use a more scientific style for the seentence "The group of authors has also published an article about...".

RESPONSE: The introduction has been enhanced as suggested. We have deleted the non-relevant information and added more details about the use of carrageenan, the advantages and limitations of this biopolymer for packaging applications, and how its use is related to the Circular Bioeconomy. Furthermore, reference 20 was more detailed for better comprehension.

#2 - Describe the equipment used for the extraction process and the conditions in section 2.2. I also suggest to merge sections 2.3 and 2.5.6. Moreover, the selection of compositions should be justified.

RESPONSE: Thank you for your recommendations. We added the equipment specification that was used for the extraction process. About merge sections 2.3 and 2.5.6 we understanding that the methodologies used for analyzing extract and films are the same. However, for better comprehension, we opted by separated the characterization of the extracts from the characterization of the films.

#3 - The results section should be divided into subsections, following a logical order, which can be based on that presented in the Experimental part.

RESPONSE: Thank you for your recommendation. As suggested, we divided the results in subsections based on experimental part.

#4 - Compare the antioxidant properties of this extract with others obtained from vegetable and fruit wastes to elucidate the potential of these ones.

RESPONSE: Thank you for your recommendation. We compared the antioxidant activity obtained in the present study with antioxidant activity from other plant materials and improved the discussion.

#5 - Include a table/graph for the antimicrobial results and improve the discussion.

RESPONSE: Thank you for your recommendation. We added a table with antimicrobial results from different concentrations of the extract analyzed and improved the discussion.

#6 Include and discuss the values of the Young modulus in the mechanical properties

RESPONSE: Thank you for your recommendation. We added the Young modulus and discussed the results.

#7 The WVP is not related to the film thickness but to material composition. Values should be compared with other biopolymer films since this material is expected to show a poor water barrier due to its high hydrophilicity.

RESPONSE: Thank you for your explanation and suggestion. We agreed and modified the discussion. We also added a comparison of the results obtained for the carrageenan films with films of other biopolymers.

#8 Explain why the extract reduced swelling but did not provide a significant effect on the water vapor performance.

RESPONSE: Thank you for your recommendation. We improved the discussion.

#9 The active properties of the films should be compared with the neat extract to determine the effect of processing and release from the film samples.

RESPONSE: Thank you for your recommendation. We compared the active properties of the films with active properties of the extract and improved the discussion.

#10 The discussion of the FTIR measurements should be focused on the interaction of the extract with the polysaccharide. 

RESPONSE: Thank you for your recommendation. We improved the discussion.

#11 - In the conclusion section authors can better explain the design of the intelligent packaging material based on pH changes. In other words, what type of food or good can be packaged so that it can present such a large changes on the pH values and then show color variations.  

RESPONSE: Thank you for your recommendation. We added a part that explains what type of food can be packaged by the material developed in conclusion, as suggested, and in the section “Results and Discussion” for better comprehension.

Reviewer 2 Report

The manuscript by Avila et al describes the development of a bioactive and intelligent film to be used as new food packaging materials. The manuscript is very interesting, it needs a deep revision.

Abstract: line 25 - authors said that the incorporation of JBE in the films improves the opacity. Is it good? Please explain.

Keywords: I suggest the addition of more keywords, for example: jaboticaba, anthocyanins, pH-sensitive...

Introduction:

Line 37 - please correct the reference [Asiakina].

Line 38 - "consumption" of synthetic plastics? I suggest: "use" of synthetic plastics.

Line 58 - Myrtaceae should be in italics.

Line 91 - Which group? Please be more specific.

Materials and Methods:

Lines 104 and 105 - The sentence "For extraction...concentration of 0.1M." does not make sense. Please correct.

Line 108 - Escherichia coli shoul be in italics. Why did the authors performed the antimicrobial studies using only E. coli (Gram-negative)? The authors considered to use a Gram-positive bacteria?

Line 119 - Why did the authors used the pH=1? 

Line 125 - I suggest that authors briefly describe the Folin-Ciocalteu method  and the DPPH free radical scavenging assay.

Line 135 - "radicals scavenged by DPPH radical", this sentence in incorrect and does not make any sense. Please verify.

Line 136 - The most accepted and employed method to determine the total anthocyanins is the pH differential method. The authors should explain why they used the described method. Moreover, the abbreviation Cn-3-Glu must be defined when it first appears.

Line 159 - Why did the authors make the incubation at 35ºC? The optimal growth temperature for E. coli is 37ºC. Why 16h of incubation? The CLSI standard protocol says 18 to 24 h. Why did the authors not studied the antimicrobial activity of the developed bioactive films?

Lines 208, 209, 233... - Please correct "biofilms" to "films". Biofilms are communities od microorganisms. 

The task "2.5.7. Statistical Analysis" should be the last task in the Materials and Methods section. It not make sense in the place as it is now.

The task "2.6. Functional groups" should follows de film preparation and not in the end of the Materials and Methods. Furthermore, this task should be name as"FTIR-ATR analysis of the films". 

Results on line 251 should be named "Results and Discussion". This entire section should be corrected according to my comments on Materials and Methods Section and should be devided in the part concerning the extract and the part related to the films. 

Table 1: What are the units of thickness and opacity?

Figure 2: How did the authors calculated the color changes (%)? 

The Conclusions should be summarized and consists only in one or two paragraphs highlighting the major results.

In addition to my above comments, the entire manuscript should be revised for grammar and spelling errors. I've detected several mistakes. 

Author Response

Promising new material for food packaging: an active and intelligent carrageenan film with natural jaboticaba additive

Author's Response

ATTENTION:

The Edıtor-In-Chıef Measurement

RESPONSES TO EDITORS’S AND REVIEWER’S COMMENTS ON MANUSCRIPT FOODS- 1415889.

Title: Promising new material for food packaging: an active and intelligent carrageenan film with natural jaboticaba additive

First of all, we would like to thank you for making your time available and kindly evaluating our manuscript. Your suggestions and comments have been valuable, thus helping to improve the quality of our manuscript. After a detailed analysis of the comments and questions, as well as the errors and suggestions contained in its opinion, the manuscript underwent some changes. We respectfully acknowledge and appreciate the invaluable contributions of the reviewer. Corrections made are highlighted in red color font. We describe below the treatment given to each of the amendments proposed by the reviewer for resubmission of the manuscript Foods-1415889.

Reviewer #2

#1 -The manuscript by Avila et al describes the development of a bioactive and intelligent film to be used as new food packaging materials. The manuscript is very interesting, it needs a deep revision.

RESPONSE: We thank the reviewer for appreciating our effect. The manuscript underwent some changes, which are indicated in the sequence.

#2 Abstract: line 25 - authors said that the incorporation of JPE in the films improves the opacity. Is it good? Please explain.

RESPONSE: Thank you for your recommendation. We added an explanation in our manuscript.

#3 Keywords: I suggest the addition of more keywords, for example: jaboticaba, anthocyanins, pH-sensitive...

RESPONSE: Thank you for your suggestion. We added more keywords.

#4 Line 37 - please correct the reference [Asiakina].

RESPONSE: Sorry, we have corrected this mistake.

#5 Line 38 - "consumption" of synthetic plastics? I suggest: "use" of synthetic plastics.

RESPONSE: Rightly so. The sentence has been modified.

#6 Line 58 - Myrtaceae should be in italics.

RESPONSE: Rightly so. We put the word in italic.

#7 Line 91 - Which group? Please be more specific.

RESPONSE: Thank you for your recommendation. We improved this sentence.

#8 Lines 104 and 105 - The sentence "For extraction...concentration of 0.1M." does not make sense. Please correct.

RESPONSE: Thank you for your recommendation. We corrected this mistake.

#9 Line 108 - Escherichia coli should be in italics. Why did the authors performed the antimicrobial studies using only E. coli (Gram-negative)? The authors considered to use a Gram-positive bacteria?

RESPONSE: Thank you for your recommendation. We put the word in italics. About microbial analysis, we understand the importance of doing a complete study, including the evaluation of the performance of the extract against Gram-positive and Gram-negative bacteria. However, due to the pandemic moment, we can not accomplish this. We consider doing a complete microbial analyze in the future.

#10 Line 119 - Why did the authors used the pH=1? 

RESPONSE: The choice of extraction parameters (such as solvent pH) was based on a previous study in which an experimental design was carried out evaluating the effect of extraction temperature and the pH of the extraction solvent. This study indicated the optimal condition for the recovery of total phenolic compounds and total anthocyanins from jaboticaba peels by the maceration technique.

#11 Line 125 - I suggest that authors briefly describe the Folin-Ciocalteu method  and the DPPH free radical scavenging assay.

RESPONSE: Thank you for your suggestion. We agree and added a brief description of the methods used.

#12 Line 135 - "radicals scavenged by DPPH radical", this sentence in incorrect and does not make any sense. Please verify.

RESPONSE: Thank you for your recommendation. We corrected this mistake.

#13 Line 136 - The most accepted and employed method to determine the total anthocyanins is the pH differential method. The authors should explain why they used the described method. Moreover, the abbreviation Cn-3-Glu must be defined when it first appears.

RESPONSE: We chose this methodology based on some authors who reported its applications, as can be seen in the following references:

  1. Svensson, D.; Svensson, D. Effects of heat treatment and additives on the anthocyanin content in blackcurrants and its relation to colour and texture Effects of heat treatment and additives on the anthocyanin content in blackcurrants and its relation to colour and texture. 2010.
  2. Sripakdee, T.; Mahachai, R.; Chanthai, S. Direct analysis of anthocyanins-rich Mao fruit juice using sample dilution method based on chromophores/fluorophores of both cyanidin-3-glucoside and pelargonidin-3-glucoside. Int. Food Res. J. 2017, 24, 215–222.
  3. Åžakar, D., KaraoÄŸlan, G.K., Gümrükçü, G. and Özgür, M.Ü. Determination of anthocyanins in some vegetables and fruits by derivative spectrophotometric method. Reviews in Analytical Chemistry, vol. 27, no. 4, 2008, pp. 235-250. https://doi.org/10.1515/REVAC.2008.27.4.235

 We have added these references to our manuscript for further clarification. About the abbreviation (Cn-3-Glu) we added a definition in the first appearance.

#14 Line 159 - Why did the authors make the incubation at 35ºC? The optimal growth temperature for E. coli is 37ºC. Why 16h of incubation? The CLSI standard protocol says 18 to 24 h. Why did the authors not studied the antimicrobial activity of the developed bioactive films?

RESPONSE: We use the parameters described by CLSI 2015 that reported the incubation temperature is 35 °C and the incubation time is 16-20h. This information can be seen in the following reference:

  1. CLSI (2015) Methods for dilution antimicrobial susceptibility tests for bacteria that grow aerobically; approved standard - tenth edition. CLSI document M07 - A10. Wayne, PA: Clinical and Laboratory Standards Institute; 2015.

The antimicrobial activity of the films can not be evaluated due to the pandemic moment. However, we consider evaluating in the future.

#15 Lines 208, 209, 233... - Please correct "biofilms" to "films". Biofilms are communities od microorganisms. 

RESPONSE: Thank you for your explanation and suggestion. We corrected this mistake.

#16 The task "2.5.7. Statistical Analysis" should be the last task in the Materials and Methods section. It not make sense in the place as it is now.

RESPONSE: Thank you for your recommendation. We agreed and modified the order of the section.

#17 The task "2.6. Functional groups" should follows de film preparation and not in the end of the Materials and Methods. Furthermore, this task should be name as"FTIR-ATR analysis of the films". 

RESPONSE: Thank you for your recommendation. We agreed and modified the order and name of the section.

#18 Results on line 251 should be named "Results and Discussion". This entire section should be corrected according to my comments on Materials and Methods Section and should be devided in the part concerning the extract and the part related to the films

RESPONSE: Thank you for your recommendation. We agreed and modified the name of the section and divided in the part concerning the extract and the part related to the films, as suggested.

#19 Table 1: What are the units of thickness and opacity?

RESPONSE: The unit of thickness is “mm” and the opacity is “Abs600nm.mm-1”. We added the units in Table 2.

#20 Figure 2: How did the authors calculated the color changes (%)? 

RESPONSE: Color changes were calculated using the same methodology described for evaluating the color property of films (section 2.5.2). However, in this case, the control was the films containing extract (50 and 100%) that had no contact with buffer solutions of different pH values.

#21 The Conclusions should be summarized and consists only in one or two paragraphs highlighting the major results.

RESPONSE: Thank you for your recommendation. We modified and improved the conclusion.

#22 In addition to my above comments, the entire manuscript should be revised for grammar and spelling errors. I've detected several mistakes. 

RESPONSE: Thank you for your observation. We revised and improved the manuscript.

Round 2

Reviewer 1 Report

Based on the author changes, this manuscript can be accepted in its current form.

Reviewer 2 Report

The authors have addressed all my questions and corrected the manuscript, whihc can now be accepted for publication.